# EllipWeather: Gaussian Ellipsoid Representation for Weather Modeling

## Abstract

Accurate weather forecasting plays a critical role in a variety of sectors, including disaster management, agriculture, transportation, and energy consumption. Most deep learning approaches for weather forecasting rely on pixel-based representations of weather data, leading to significant data redundancy and inefficiencies in capturing the weather's continuous and dynamic nature. To address these challenges, we propose a novel approach, **EllipWeather**, which leverages Gaussian ellipsoids to represent weather patterns, as weather phenomena can be effectively modeled using a mixture of Gaussian distributions. With this representation, we first develop an equivariant graph neural network to capture the intrinsic equivariance of weather variances, specifically tailored to process Gaussian ellipsoids for weather prediction tasks. Then we also demonstrate the potential of EllipWeather in downstream tasks such as data compression and downscaling (super-resolution). Extensive experiments on widely used datasets show that EllipWeather achieves superior performance over previous works.

## 1 Introduction

Accurate weather modeling is fundamental to modern society, profoundly impacting a wide array of sectors, including disaster management (Merz et al., 2020), agriculture (Bendre et al., 2015), transportation (Dey et al., 2014), and energy consumption (Meenal et al., 2022). At its core, weather forecasting involves predicting the state of the atmosphere at a specific time and location (Fathi et al., 2022), providing the critical information needed to mitigate risks and facilitate informed decision-making across these domains.

Data-driven models have revolutionized weather forecasting, with models like GraphCast (Lam et al., 2023) and Aurora (Bodnar et al., 2025) achieving state-of-the-art perfromance that exceeds traditional Numerical Weather Prediction (NWP) systems at a fraction of the computational cost. However, both data-driven and physics-based models are fundamentally constrained by their reliance on a discretized, grid-based (i.e., pixel-level) representation of the atmosphere. This paradigm of modeling a continuous fluid system on a discrete grid creates a critical representation bottleneck, leading to significant challenges in computational efficiency, physical realism, and data handling.

The limitation of the grid-based representation is twofold. First, it leads to massive data volumes, creating significant burdens for data storage, transmission, and processing (Brotzge et al., 2023; Shi et al., 2025). Weather datasets are often terabytes in size, and the high resolution required for accurate forecasts exacerbates this issue. Second, the rigid grid structure fails to capture the inherently continuous and fluid nature of weather phenomena (Bonavita, 2024). Weather systems are dynamic and evolve smoothly over space and time, yet a grid-based approach represents both areas of intense activity and calm conditions with the same high resolution. This can result in predictions that are spatially redundant or prone to artifacts, and it makes tasks like resolution enhancement (super-resolution) non-trivial (Harder et al., 2023; Gruca et al., 2023).

To tackle these problems, we propose a paradigm shift from a discrete, pixel-based representation to a continuous, object-based one. Instead of modeling weather as a field of pixel values, we propose **EllipWeather** to represent it as a collection of continuous, physically meaningful gaussian **ellip**soids as **weather** phenomena can be effectively modeled using a mixture of Gaussian distributions (Wang et al., 2015). In EllipWeather, a weather variable is no longer described by thousands of pixels but by a collection of few Gaussian ellipsoids, which can be described by parameters including a mean

vector to represent its location, a scaling factor to represent its shape, a rotation vector to represent its orientation, and a weight to represent its intensity. This provides a compact, continuous, and resolution-agnostic representation of the weather state.

Building on this representation, we construct a group equivariant graph neural network (EGNN) that operates directly on the set of Gaussian ellipsoids. Each Gaussian ellipsoid is taken as a node and edges are formed dynamically based on Euclidean distance, enabling adaptive relational reasoning without committing to a rigid Eulerian grid. The EGNN updates both feature embeddings and refined centers while preserving rotational and translational consistency.

Beyond forecasting, we also show the potential of EllipWeather for compression, *i.e.*, storing O(K) ellipsoid parameters instead of O(HW) pixels while retaining reconstruction fidelity,resolution-agnostic downscaling (super-resolution), *i.e.*, rendering ellipsoids onto arbitrary target grids without retraining, and interpretable diagnostics, *i.e.*, each component corresponds to a physically meaningful mesoscale entity.

The main contributions of our work are as follows:

- We propose a novel weather representation using Gaussian ellipsoids, which is more compact and efficient than traditional pixel-based methods.
- We design an equivariant graph neural network capable of processing Gaussian ellipsoids, capturing the intrinsic equivariance of weather variances for forecasting tasks.
- We demonstrate the potential of this representation in downstream tasks such as data compression and super-resolution.

## 2 RELATED WORK

### 2.1 DEEP LEARNING FOR WEATHER FORECASTING

Data-driven weather forecasting has seen rapid progress, with models like FourCastNet (Pathak et al., 2022), GraphCast (Lam et al., 2023), Aurora (Bodnar et al., 2025) and so on achieving performance that rivals or exceeds traditional Numerical Weather Prediction systems while being significantly more computationally efficient. These models typically employ convolutional neural networks (CNNs) or graph neural networks (GNNs) to capture spatial and temporal dependencies in weather data. However, they build on pixel-based representations, which can lead to inefficiencies in capturing the continuous nature of weather phenomena. Our work departs from this grid-based paradigm, proposing an object-based representation to directly address these challenges.

### 2.2 EQUIVARIANT NEURAL NETWORKS FOR WEATHER MODELING

Forcing a neural network to respect the fundamental symmetries of a physical system is a powerful inductive bias. Group equivariant neural networks, which are equivariant to group operations, like rotations, translations, and reflections, are designed to preserve these geometric properties by construction (Suk et al., 2023; Xu et al., 2023). By building these symmetries directly into the network architecture, models can achieve higher data efficiency and learn more physically plausible dynamics (Satorras et al., 2021). Pioneering explorations led to the development of Spherical CNNs, which generalize convolutions to the spherical domain using spherical harmonics (Cohen et al., 2018; Esteves et al., 2018). The Fourier basis in FourCastNet naturally handles spherical periodicity, while the icosahedral mesh of GraphCast provides a more uniform discretization of the globe amenable to periodic equivariant message passing (Pathak et al., 2022; Lam et al., 2023). EllipWeather builds on this line of work by applying group equivariant GNN not to a grid or a set of particles, but to a collection of parametric ellipsoid objects representing weather phenomena.

### 2.3 PRIMITIVE-BASED REPRESENTATIONS FOR WEATHER DATA

Representing images or scenes using a set of geometric primitives has been explored in computer vision and graphics (Hu et al., 2025; Zhang et al., 2024). Inspired by recent breakthroughs in computer vision like 3D Gaussian Splatting (3DGS) (Kerbl et al., 2023), Wang et al. adapted 3DGS to represent dynamic scenes as a collection of anisotropic 3D Gaussians, and paired with Mamba

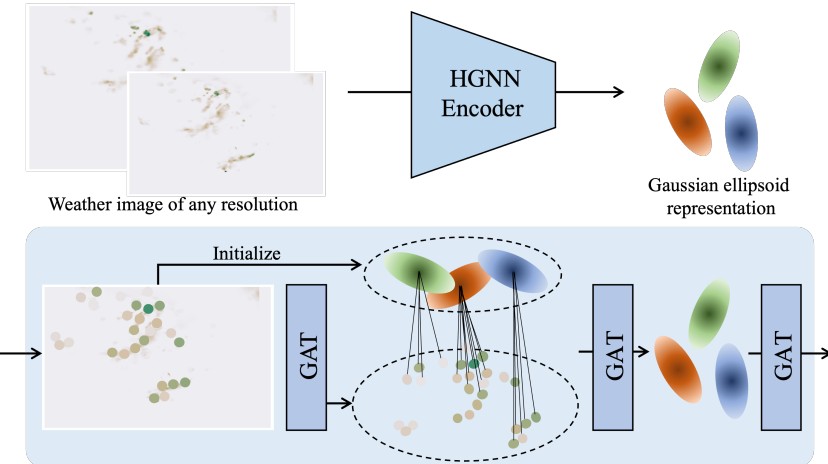

Figure 1: Hierarchical graph neural encoder for EllipWeather. Top row: encoding an image into the ellipsoid space; bottom row: HGNN architecture.

(Gu & Dao, 2023) for prediction. While this marks a significant advance, the prediction model itself lacks explicit physical inductive biases. Our EllipWeather adopts a similar strategy of representing the weather field as a collection of geometric primitives but introduce a crucial innovation in the prediction methodology. We replace the general sequence model with a group-equivariant Graph Neural Network. This allows our model to learn the interactions and temporal evolution of the primitives in a manner that explicitly respects fundamental physical symmetries, offering a more principled and robust framework for dynamics modeling on these modern, efficient representations. Besides, we also propose a one-step gaussian fitting algorithm to convert pixel-based weather data into our proposed Gaussian ellipsoid representation and show its effectiveness in data compression and super-resolution tasks.

## 3 METHODOLOGY

We first introduce the Gaussian ellipsoid representation for weather data. Then the design of an equivariant graph neural network tailored for weather forecasting tasks. Finally, we discuss how to leverage the proposed representation for downstream tasks, data compression and super-resolution.

### 3.1 PRELIMINARY

A weather variable (e.g., temperature, humidity, wind velocity) can be described per-pixel on a 2D grid as a matrix $X \in \mathbb{R}^{H \times W}$, where $H$ and $W$ are the height and width of the grid, respectively. Each element $X_{ij}$ represents the value of the weather variable at the grid cell located at row $i$ and column $j$. We focus on a single weather variable and 2D spatial domain for simplicity. Without loss of generality, our method can be extended to multiple variables and 3D spatial domains.

### 3.1.1 GAUSSIAN ELLIPSOID

We represent a weather variable matrix $X$ with a set of $K$ Gaussian ellipsoids $G$, where each ellipsoid $G_k = \{\mu_k, a_k, r_k, w_k\}$ is defined by a location vector $\mu_k \in \mathbb{R}^2$, a scaling factor $a_k \in \mathbb{R}^2$, a rotation vector $r_k \in \mathbb{R}^3$, and weight $w_k \in \mathbb{R}$. The grid value at location $(i, j)$ can be reconstructed from the Gaussian ellipsoids as:

$$\hat{X}_{ij} = \sum_{k=1}^{K} w_k \cdot \exp\left(-\frac{1}{2}(p_{ij} - \mu_k)^T R_k A_k^{-1} R_k^T (p_{ij} - \mu_k)\right), \quad (1)$$

where $p_{ij} = [i, j]^\top$, $A_k = \text{diag}(a_k)$, and $R_k$ are the coordinate of grid cell $(i, j)$, scaling matrix, and rotation matrix derived from the rotation vector $r_k$ using Rodrigues' rotation formula (Rodrigues, 1840) respectively.

### 3.1.2 Equivariance for Weather Variances

Weather phenomena exhibit intrinsic symmetries, particularly rotational and translational equivariance. This means that if the input weather data is rotated or translated, the output predictions should undergo the same transformation. Formally, let $T_g$ be a transformation operator corresponding to a group element $g$ (e.g., rotation or translation). A function $f$ is equivariant with respect to the group $G$ if:

$$f(T_g(X)) = T_g(f(X)), \quad \forall g \in G \tag{2}$$

In the context of weather forecasting, this property ensures that the model's predictions remain consistent under spatial transformations, which is crucial for accurately modeling the dynamics of weather phenomena.

### 3.1.3 Problem Formulation

**Gaussian Ellipsoid Fitting.** To obtain the Gaussian ellipsoid representation from pixel-based weather data, the Gaussian ellipsoid fitting problem can be formulated as an optimization problem:

$$\min_{\{G_k\}_{k=1}^K} \sum_{i=1}^H \sum_{j=1}^W \left( X_{ij} - \hat{X}_{ij} \right)^2 + \lambda \sum_{k=1}^K \|w_k\|^2 \tag{3}$$

where $\hat{X}_{ij}$ is obtained through Eq. 1 and $\lambda$ is a regularization parameter. The goal is to find the set of Gaussian ellipsoids that best reconstruct the original weather variable matrix $X$.

**Equivariant Weather Forecasting.** Given a sequence of weather variable matrices $\{X_t\}_{t=1}^T$, the goal of weather forecasting is to predict the future state $X_{T+1}$ based on the past observations. Using the Gaussian ellipsoid representation, this can be reformulated as predicting the future set of Gaussian ellipsoids $\{G_k^{T+1}\}_{k=1}^K$ from the past sets $\{\{G_k^t\}_{k=1}^K\}_{t=1}^T$. The forecasting model $f$ should be equivariant to spatial transformations:

$$\{G_k^{T+1}\}_{k=1}^K = f(\{\{G_k^t\}_{k=1}^K\}_{t=1}^T) \tag{4}$$

### 3.2 Gaussian Ellipsoid Representation

We aim to fit a set of $K$ Gaussian ellipsoids $G = \{G_k\}_{k=1}^K$ to approximate the weather variable matrix $X$. We use two ways to get the Gaussian ellipsoid representations: (1) following Wang et al., we can use an iterative fitting algorithm that starts with an initial guess of the Gaussian ellipsoids and iteratively refines them to minimize the reconstruction error. (2) We also propose a hierarchical graph neural network (HGNN) to predict the Gaussian ellipsoid parameters directly from the pixel-based weather data as shown in Fig. 1. HGNN takes the weather variable matrix $X$ of any resolutions as input and outputs the parameters of $K$ Gaussian ellipsoids. The network is trained to minimize the reconstruction error between $X$ and $\hat{X}$, where $\hat{X}$ is obtained by rendering the predicted Gaussian ellipsoids as Eq. 1.

**Initialization.** Given an input weather variable matrix $X$, we first represent it as a graph $G^p = (V^p, E^p)$, where each pixel corresponds to a node $v_i^p \in V^p$ with feature $x_i^p = X_{ij}$, and edges $e_{ij}^p \in E^p$ are formed based on spatial proximity. Then we initialize $K$ Gaussian ellipsoids by randomly sampling $K$ nodes from the graph as their initial centers $\mu_k$. The initial scaling factors $a_k$ are set to a constant value, the rotation vectors $r_k$ are initialized as zero vectors, and the weights $w_k$ are initialized based on the pixel values of the sampled nodes. Since weather matrices are represented as graphs, our method can handle input matrices of any resolutions.

**Hierarchical Graph Neural Network.** HGNN intends to learn to represent a group of pixels as a mixture of Gaussian distributions. With the initialized Gaussian ellipsoids $G$ and the pixel graph $G^p$, we first perform message passing on the pixel graph to update the pixel features, allowing each pixel to aggregate information from its neighbors through graph attention layers (GAT) (Veličković et al., 2018). Then we construct a bipartite graph $G^b = (V^p, V^g, E^b)$ between the pixel nodes $V^p$ and the Gaussian ellipsoid nodes $V^g$. Edges $e_{ik}^b \in E^b$ are formed dynamically based on Euclidean distance between pixel $v_i^p$ and the center of Gaussian ellipsoid $v_k^g$. GAT layers are also used here for message

passing on the bipartite graph to update the Gaussian ellipsoid features by aggregating information from the connected pixel nodes. Finally, we construct a Gaussian ellipsoid graph $G^g = (V^g, E^g)$, where each Gaussian ellipsoid corresponds to a node $v_k^g \in V^g$ with feature $x_k^g = [\mu_k, a_k, r_k, w_k]$, and edges $e_{kl}^g \in E^g$ are formed dynamically based on Euclidean distance between the centers of the Gaussian ellipsoids. We perform message passing on the Gaussian ellipsoid graph to update the Gaussian ellipsoid features by aggregating information from neighboring ellipsoids. The updated features are then used to predict the final parameters of the Gaussian ellipsoids.

**Training Objective.** To train HGNN, we minimize the reconstruction error between the original weather variable matrix $X$ and the reconstructed matrix $\hat{X}$ obtained by rendering the predicted Gaussian ellipsoids using Eq. 1.

## 3.3 EQUIVARIANT GRAPH NEURAL NETWORK FOR WEATHER FORECASTING

Based on the EllipWeather representation, the variation of weather can be modeled as the temporal evolution of a set of Gaussian ellipsoids. We design an equivariant graph neural network (EGNN) to capture the intrinsic equivariance of weather variances for forecasting tasks.

### 3.3.1 MODEL ARCHITECTURE

Our model is designed to be equivariant with respect to the 2D Euclidean group $E(2) = SO(2) \ltimes \mathbb{R}^2$. Following the principle of E(n)GNN (Satorras et al., 2021), we ensure that each update rule either depends only on group invariants or transforms consistently with the group action applied to its inputs. We consider four types of latent variables: location $\mu$, rotation $r$, scaling $a$, and weight $w$.

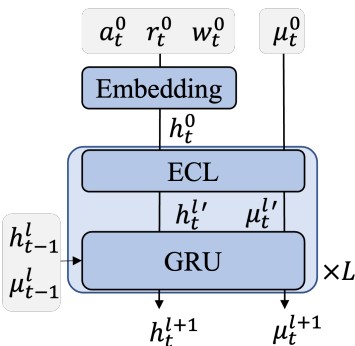

Each update module consists of a equivariant convolutional layer (EGL) and a gated recurrent unit as shown in Fig. 2. Equivariance in our design goes beyond coordinates. The location $\mu$ naturally transforms under translations and rotations. For the rotation variable $r$, if the system is rotated by $\phi$, its local orientation should also shift by $\phi$; otherwise the representation becomes inconsistent with the global frame. The scaling variable $a$ is not meaningful without an orientation, since $(a, r)$

Figure 2: Illustration of the equivariant update module.

together define anisotropic covariance structures that must rotate coherently under global transformations. Finally, $w$ is scalar-valued and thus invariant. By ensuring that $(\mu, r, a)$ transform equivariantly while $w$ remains invariant, the model maintains geometric consistency under $E(2)$ actions. This design enables robust generalization to translated and rotated sequences.

### 3.3.2 EQUIVARIANT UPDATE RULES

We take the first layer for illustration. Given the input Gaussian ellipsoid parameters at time step $t$ and layer 0, $G_t^{l=0} = \{a_t^{l=0}, r_t^{l=0}, w_t^{l=0}, \mu_t^{l=0}\}$, we describe the update rules for each latent variable as follows.

**Embedding layer.** We separate the input parameters into two parts: the positional part $\mu_t^{l=0}$ and the non-positional part $\{a_t^{l=0}, r_t^{l=0}, w_t^{l=0}\}$. For the non-positional part, we embed it into a high-dimensional feature space using an MLP as:

$$h_t^{l=0} = \text{MLP}([\log a_t^{l=0}, \cos \|r_t^{l=0}\|, \sin \|r_t^{l=0}\|, w_t^{l=0}]),\tag{5}$$

where $h_t^{l=0}$ is the hidden feature vector, please refer to App. A.2 for more details.

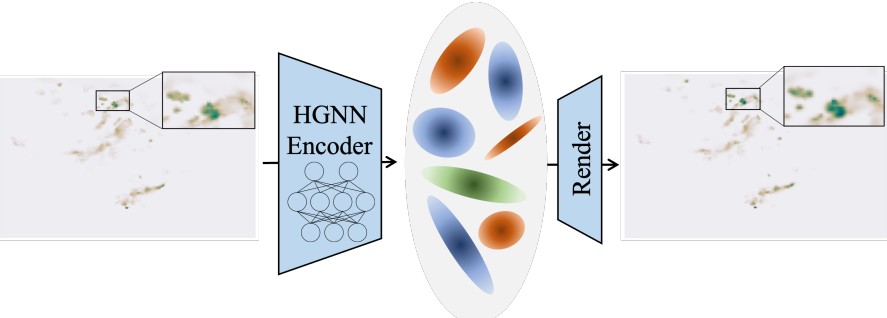

Figure 3: The grid-based weather matrix is encoded into Gaussian Ellipsoid space through a neural network HGNN. The ellipsoid parameters can then be decoded back to the grid space through the render. This process allows for (1) data compression by encoding weather data into the Gaussian Ellipsoid space and (2) super-resolution by reconstructing a higher-resolution weather matrix from the ellipsoid representation.

**Equivariant convolutional layer (ECL).** The ECL updates both th hidden features and the position vectors as:

$$\mu_{t,i}^{l'} = \mu_{t,i}^{l=0} + \sum_{j \in \mathcal{N}(i)} \phi_\mu(h_{t,i}^{l=0}, h_{t,j}^{l=0}, \|\mu_{t,i}^{l=0} - \mu_{t,j}^{l=0}\|^2)\, (\mu_{t,i}^{l=0} - \mu_{t,j}^{l=0}), \tag{6}$$

$$h_{t,i}^{l'} = \phi_h\big(h_{t,i}^{l=0}, \sum_{j \in \mathcal{N}(i)} \phi_\mu(h_{t,i}^{l=0}, h_{t,j}^{l=0}, \|\mu_{t,i}^{l=0} - \mu_{t,j}^{l=0}\|^2)\big), \tag{7}$$

where $\mathcal{N}(i)$ is the set of neighbors of node $i$, $\phi_\mu(\cdot)$ and $\phi_h(\cdot)$ are MLPs.

**Gated recurrent unit (GRU).** For temporal prediction, we introduce group-consistent gates for hidden features and position vectors respectively. Each gate is computed only from $E(2)$-invariant scalars, ensuring that the update preserves equivariance. The GRU updates both the hidden features and the position vectors as:

$$\mu_{t,i}^{l+1} = (1 - g_{t,i}^\mu)\, \mu_{t,i}^{l'} + g_{t,i}^\mu\, \mu_{t-1,i}^l, \tag{8}$$

$$h_{t,i}^{l+1} = (1 - g_{t,i}^h)\, h_{t,i}^{l'} + g_{t,i}^h\, h_{t-1,i}^l, \tag{9}$$

where $g_{t,i} \in (0,1)$ is the gate computed from invariant features $\psi_t = \big(\|\mu_t - \overline{\mu_t}\|_2, \|\mu_{t-1} - \overline{\mu_{t-1}}\|_2, \|\mu_t - \mu_{t-1}\|_2\big)$ through two different MLPs, $\sigma(\phi_{gh}(\psi_t))$ and $\sigma(\phi_{g\mu}(\psi_t))$ respectively.

**Output layer.** Through $L$ layers of ECL and GRU, we obtain the final hidden features $h_t^L$ and position vectors $\mu_t^L$. Then predict the updates for the non-positional variables as:

$$\Delta r_t = \phi_r(h_t^L), \quad \Delta \log a_t = \phi_a(h_t^L), \quad \Delta w_t = \phi_w(h_t^L), \tag{10}$$

where $\phi_r(\cdot)$, $\phi_a(\cdot)$ and $\phi_w(\cdot)$ are MLPs. Finally, we have the variables of next time step $t+1$ as:

$$\mu_{t+1} = \mu_t^L, \tag{11}$$

$$r_{t+1} = r_t^{l=0} + \gamma_t\, \Delta r_t, \tag{12}$$

$$\log a_{t+1} = \log a_t^{l=0} + \eta_t\, \Delta \log a_t, \tag{13}$$

$$w_{t+1} = w_t^{l=0} + \zeta_t\, \Delta w_t, \tag{14}$$

where $\gamma_t, \eta_t, \zeta_t \in (0,1)$ are gates computed from the same invariant feature set $\psi_t$.

**Training Objective.** To train the EGNN, we minimize the forecasting error directly on the Gaussian ellipsoid parameters. Specifically, we compute the loss as the difference between the ground truth Gaussian ellipsoid parameters $\{G_k^{t+1}\}_{k=1}^K$ and the predicted parameters $\{\hat{G}_k^{t+1}\}_{k=1}^K$, ensuring that the model learns to predict the evolution of the ellipsoids accurately.

Without loss of generality, we can perform EGNN on both 2D and 3D Gaussian ellipsoids.

Table 1: Experiments on temporal forecasting. The arrows indicate whether higher (↑) or lower (↓) values are better. Best results are in **bold**, and second best are underlined.

| Method | MAE ↓ | SSIM ↑ | LPIPS ↓ | LPIPS-radar ↓ | CSI-20 ↑ | CSI-30 ↑ | CSI-40 ↑ |
|---|---|---|---|---|---|---|---|
| ConvGRU | 0.006 | 0.819 | 0.205 | 1.621 | 0.306 | - | - |
| PhyDNet | 0.017 | 0.373 | 0.320 | 2.058 | 0.311 | 0.089 | 0.002 |
| SimVP | 0.066 | 0.379 | 0.481 | 2.925 | 0.085 | 0.088 | 0.018 |
| DiffCast | 0.157 | 0.004 | 0.932 | 4.057 | 0.049 | 0.021 | 0.021 |
| Mamba | 0.004 | 0.899 | 0.129 | 0.699 | 0.309 | 0.165 | 0.074 |
| GauMamba | **0.003** | 0.907 | 0.122 | 0.600 | 0.361 | 0.205 | 0.089 |
| **Ours** | **0.003** | **0.912** | **0.118** | **0.538** | **0.404** | **0.258** | **0.148** |

## 3.4 DATA COMPRESSION & SUPER-RESOLUTION

Besides weather forecasting, the proposed Gaussian ellipsoid representation also enables downstream tasks such as data compression and super-resolution as shown in Fig. 3.

**Data Compression.** Instead of storing the full pixel-based weather data, we can store the parameters of the Gaussian ellipsoids. Given a weather variable matrix $X \in \mathbb{R}^{H \times W}$, we can fit $K$ Gaussian ellipsoids to it and store their parameters $\{G_k\}_{k=1}^K$. The compression ratio can be calculated as: Compression Ratio $= \frac{H \times W}{K \times P}$ where $P$ is the number of parameters per Gaussian ellipsoid (7 for 2D: 2 for location, 2 for scaling, 3 for rotation, and 1 for weight). By choosing an appropriate $K$, we can achieve significant compression while retaining reconstruction fidelity.

**Super-Resolution.** The Gaussian ellipsoid representation allows for resolution-agnostic rendering of weather data. Given a set of Gaussian ellipsoids $\{G_k\}_{k=1}^K$, we can render them onto any target grid size $H' \times W'$ using Eq. 1. This means that we can generate high-resolution weather variable matrices from a low-resolution input by fitting Gaussian ellipsoids to the low-resolution data and then rendering them onto a higher-resolution grid. This approach enables super-resolution without retraining the model, as the rendering process is independent of the grid resolution.

## 4 EXPERIMENTS

**Setup.** We first introduce the datasets and evaluation metrics used in our experiments. Then we present the experimental results on weather forecasting, data compression, and super-resolution tasks. For the **Datasets**, we evaluate our method on two widely used weather datasets, NEXRAD (Department of Atmospheric Sciences, Texas A&M University & School of Meteorology, University of Oklahoma, 2021) and RainNet (Chen et al., 2022). *NEXRAD* consists of radar observations of major storms across the United States, including 6,255 frames of 3D radar observations with a grid size of $512 \times 512 \times 44$ and interval of 5 minites. Each voxel contains 7 radar features. *RainNet* contains 62,400 pairs of high-quality low/high-resolution precipitation maps from 2002 to 2018 with an interval of 1 hour. The sizes of the high-resolution and low-resolution precipitation map are $624 \times 999$ and $208 \times 333$, respectively. For the **Evaluation Metrics**, following previous works (Pathak et al., 2022; Lam et al., 2023; Chen et al., 2022; Wang et al.), we mainly employ Mean Squared Error (MSE), Mean Absolute Error (MAE), and Critical Success Index (CSI) to evaluate the forecasting performance. For data compression and super-resolution tasks, we use Peak Signal-to-Noise Ratio (PSNR) and Structural Similarity Index Measure (SSIM) as evaluation metrics.

## 4.1 EXPERIMENTAL RESULTS

We first evaluate the performance of our method on temporal forecasting. Then we demonstrate the potential of our Gaussian ellipsoid representation for data compression and super-resolution tasks.

### 4.1.1 TEMPORAL FORECASTING RESULTS

For temporal forecasting, we mainly compare our method with ConvGRU Shi et al. (2017), PhyDNet Guen & Thome (2020), SimVP Gao et al. (2022), DiffCast (Yu et al., 2024), Mamba (Zhu et al., 2024), and STC-GS (Wang et al.) on NEXRAD and RainNet datasets.

Table 2: Ablation study on the effectiveness of equivariant design in our EGNN. $e_\mu$ denotes the mean squared error of the predicted ellipsoid centers.

| Equivariance | $e_\mu\downarrow$ | MAE $\downarrow$ | MSE $\downarrow$ | SSIM $\uparrow$ | LIPS $\downarrow$ |
|---|---|---|---|---|---|
| | 19.19 | 0.00266 | 0.0004 | 0.911 | 0.107 |
| ✓ | 3.77 | 0.00168 | 0.0002 | 0.950 | 0.056 |

The quantitative results are summarized in Table 1. Our method outperforms all baselines across all metrics, demonstrating the effectiveness of the proposed Gaussian ellipsoid representation and the equivariant graph neural network for weather forecasting tasks. Notably, our method achieves significant improvements in MAE and SSIM, indicating better accuracy and structural preservation in the forecasts. The improvements in LPIPS and CSI metrics further highlight the model's ability to capture perceptual quality and critical weather events.

Besides experiments on NEXRAD dataset, we also conduct experiments on RainNet dataset, where the data is more sparse and the weather patterns are global. For MAE metric, our method achieves 0.003 which is the same as the best baseline STC-GS. For RMSE metric, our method achieves 0.015 which is slightly better than STC-GS (0.016). Our method also outperforms STC-GS on SSIM (0.975 vs 0.972) metric. Our EGNN and STC-GS models perform comparably on the RainNet dataset since the weather patterns in RainNet exhibit more global characteristics and rainfall shows stronger location-specific associations.

### 4.1.2 ABLATION STUDIES

To validate the effectiveness of equivariance design in our EGNN, we conduct ablation studies on the NEXRAD dataset by removing the equivariant constraints and replacing the EGNN with a standard GNN. We randomly rotate the input sequences during testing with angles of $\{30°, 60°, 90°, 120°, 180°\}$, and evaluate the performance of both the full model and the ablated model as shown in Table 2 . We observe a significant drop in performance when equivariance is removed, confirming the importance of incorporating physical symmetries into the model architecture for weather forecasting tasks. Furthermore, any other physical dynamics can also be incorporated into our EGNN framework, such as conservation laws and incompressibility constraints, which we leave for future work. For more details, please refer to App. A.2.

### 4.1.3 DATA COMPRESSION & SUPER-RESOLUTION RESULTS

We also show the potential of our EllipWeather representation for data compression and super-resolution tasks on the RainNet dataset.

**Data Compression.** We fit different numbers (512, 1024, 2048) of Gaussian ellipsoids to the original high-resolution precipitation maps and

Table 3: Data compression and reconstruction results with different numbers of ellipsoids.

| # Ellipsoids | PSNR $\uparrow$ | RMSE $\downarrow$ | SSIM $\uparrow$ |
|---|---|---|---|
| 512 | 56.6275 | 0.001256 | 0.9929 |
| 1024 | 57.1933 | 0.001251 | 0.9943 |
| 2048 | 57.5952 | 0.001241 | 0.9952 |

evaluate the reconstruction quality using PSNR and SSIM metrics. As shown in Table 3, our method achieves high reconstruction quality even with a small number of Gaussian ellipsoids, demonstrating its effectiveness for data compression. For instance, using only 512 Gaussian ellipsoids (compression ratio = 22.62), we achieve a PSNR of 56.63 and an SSIM of 0.9929, which is satisfing for many applications Salomon (2002); Wang et al. (2004). With the increase of the number of Gaussian ellipsoids, the reconstruction quality further improves, indicating that our representation can effectively capture the essential features of the weather data with a compact set of parameters.

**Downscaling (Super-resolution).** We conduct downscaling experiments on two settings, paired-training and unpaired-training. In the paired-training setting, we input low-resolution precipitation maps and train our model to predict the parameters of Gaussian ellipsoids by minimizing the reconstruction error between the high-resolution ground truth and the rendered high-resolution maps from the predicted Gaussian ellipsoids. In the unpaired-training setting, we input both high-resolution and low-resolution precipitation maps during training, and train our model to reconstruct the input maps by minimizing the reconstruction error. Our model is suppposed to learn a shared representation for both high-resolution and low-resolution maps. During testing, we encode low-resolution maps into Gaussian ellipsoids and render them onto high-resolution grids.

Table 4: Results on paired downscaling

| # Ellipsoids | PSNR ↑ | RMSE ↓ | SSIM ↑ |
|---|---|---|---|
| 1024 | 48.7744 | 0.001273 | 0.9904 |
| 2048 | 49.5623 | 0.001280 | 0.9934 |

Table 5: Results on unpaired downscaling

| # Ellipsoids | PSNR ↑ | RMSE ↓ | SSIM ↑ |
|---|---|---|---|
| 1024 | 47.4491 | 0.001356 | 0.9860 |
| 2048 | 48.7268 | 0.001246 | 0.9906 |

Figure 4: A comparison of our method against a non-equivariant baseline. The first row shows the ground truth, followed by the predictions of our method (Row 2) and a non-equivariant method (Row 3). The final two rows show the predictions from our method and the non-equivariant method, respectively, on a 180° rotated input. Please refer to App. C.2 for more details.

## 5 DISCUSSION

**Why we need equivariance?**   Equivariance to spatial transformations is a critical inductive bias for weather forecasting, as the underlying physical phenomena exhibit inherent symmetries. By enforcing this property in our model architecture, we ensure its predictions remain consistent when the input is transformed. This is visualized in Fig. 6, which compares our model to a non-equivariant baseline. For an input sequence rotated by 180°, the baseline model fails, producing a distorted and inaccurate forecast. Our equivariant model, however, generates a prediction that is correctly rotated, demonstrating its ability to capture the system's true dynamics. This result underscores that incorporating physical symmetries is essential for achieving robust and reliable weather forecasts.

**Advantages of EllipWeather representation.**   In addition to enabling equivariant modeling, the EllipWeather representation offers advantages over traditional pixel-based methods. First, it provides a more compact representation of weather data, allowing for efficient storage and transmission. Second, the continuous nature of Gaussian ellipsoids enables smooth interpolation and extrapolation, which is beneficial for tasks like super-resolution. Third, the parametric form of the representation facilitates the incorporation of physical constraints and domain knowledge directly into the model. Furthermore, by representing weather phenomena as ellipsoids, weather data of any resolution can be effectively used as input, providing flexibility in handling diverse datasets from different sources, which will be our future work. *For additional visual results, please refer to App. C.*

## 6 CONCLUSION

In this paper, we introduced EllipWeather, a novel framework for weather forecasting that leverages a Gaussian ellipsoid representation and an equivariant graph neural network. Our approach addresses the limitations of traditional pixel-based methods by providing a more compact and efficient representation of weather data, while also incorporating physical symmetries through equivariant modeling. We demonstrated the effectiveness of our method on benchmark datasets, achieving state-of-the-art performance in weather forecasting tasks. Additionally, we showcased the potential of our representation for downstream applications such as data compression and super-resolution. Future work will explore the integration of additional physical constraints and the extension of our framework to multi-source data.

## REPRODUCIBILITY STATEMENT

We have made extensive efforts to ensure the reproducibility of our work. Detailed dataset descriptions are provided in App. B.1, training configurations and hyperparameters are reported in App. B.2, and method details in App. A. Upon acceptance, we will release our models, together withtraining and inference code, to facilitate replication and further research.

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

## A  METHOD DETAILS

### A.1  GAUSSIAN ELLIPSOID FITTING ALGORITHM

**Iterative Fitting Algorithm.**  For the iterative fitting algorithm, we follow Wang et al. to fit a set of $K$ Gaussian ellipsoids to the weather variable matrix $X$. The algorithm starts with an initial guess of the Gaussian ellipsoids and iteratively refines them to minimize the reconstruction error. In each iteration, we perform the following steps: 1. Compute the reconstruction $\hat{X}$ using the current set of Gaussian ellipsoids. 2. Calculate the reconstruction error between $X$ and $\hat{X}$. 3. Update the parameters of the Gaussian ellipsoids using gradient descent to minimize the reconstruction error. The process is repeated until convergence or a maximum number of iterations is reached. please refer to Wang et al. for more details.

**Hierarchical Graph Neural Network.**  We provide more details about the architecture of HGNN. For the bipartite graph construction, we connect each pixel node to its $M$ nearest Gaussian ellipsoid nodes based on Euclidean distance. For the Gaussian ellipsoid graph construction, we connect each Gaussian ellipsoid node to its $N$ nearest neighbors. We use 3 GAT layers for message passing on the pixel graph, 3 GAT layers for message passing on the bipartite graph, and 3 GAT layers for message passing on the Gaussian ellipsoid graph. The hidden dimension of all GAT layers is set to 64. We use ReLU as the activation function and apply layer normalization after each GAT layer. The network is trained using the Adam optimizer with a learning rate of 0.001 and a batch size of 16. For the bipartite graph update mechanism, we illustrate it in Fig. 5. We update the edges dynamically based on the updated centers of the Gaussian ellipsoids after each message passing step.

**Equivariance for Weather Variances.**  We illustrate the concept of equivariance in Fig. 6. When the input weather data is rotated or translated with an operation $\delta$, the output predictions should undergo the corresponding transformation $\delta'$. This property ensures that the model's predictions remain consistent under spatial transformations, which is crucial for accurately modeling the dynamics of weather phenomena.

### A.2  EQUIVARIANCE DETAILS

**Clarification on non-positional embedding.**  It is important to emphasize that the embedding of non-positional parameters $(a, r, w)$ into $h$ is not required to be strictly $E(2)$-invariant. Specifically:

1. the weight $w$ is a scalar and thus invariant under any $E(2)$ action;

2. the rotation $r$ and scaling $a$ are not invariants by themselves, but jointly define the anisotropic covariance $\Sigma(a, r) = R(r) \operatorname{diag}(a^2) R(r)^\top$, which remains equivariant under global rotations;

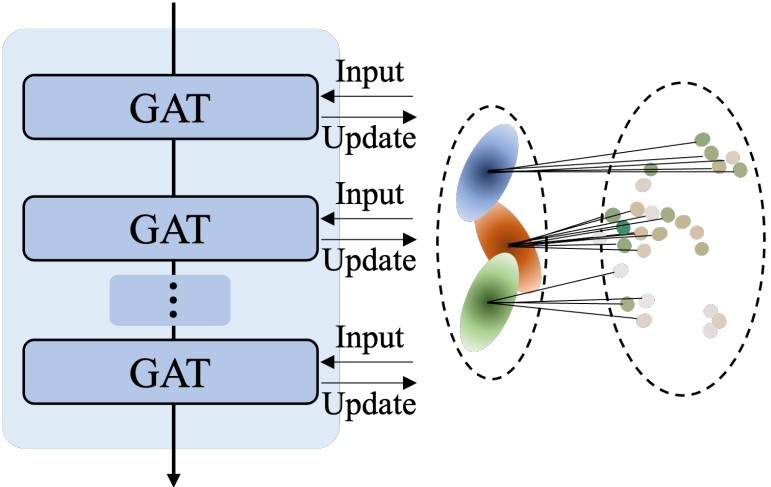

Figure 5: Bipartite graph update mechanism in EllipWeather.

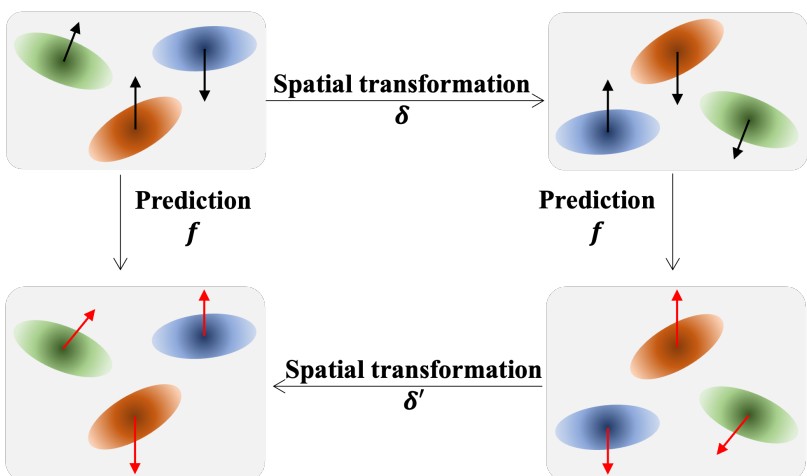

Figure 6: Illustration for equivariance.

3. in our design, the MLP embedding $h = \mathrm{MLP}([a, r, w])$ only serves as a high-dimensional feature representation. The equivariance property is ensured by the update rules: the coordinate update depends only on $E(2)$-invariant quantities, while the updates of $r$ and $a$ are defined in an explicitly equivariant manner (e.g., $r \mapsto r + \Delta r$, $\log a \mapsto \log a + \Delta \log a$).

Therefore, although $h$ itself is not strictly $E(2)$-invariant, the overall EGNN module remains rigorously $E(2)$-equivariant due to the symmetry-preserving update rules.

### A.2.1    EQUIVARIANCE

**Lemma A.1 (Gate invariance).**    Let $g = (R, t) \in E(2)$ act on $\mu_t$ by $\mu_t \mapsto R\mu_t + t$. Then each component of $\psi_t$ is invariant: $\|\mu_t\|_2 \mapsto \|R\mu_t + t\|_2$ depends on a choice of origin; using centered coordinates (i.e., relative to a global origin or mean) removes $t$, and $\|R\mu\|_2 = \|\mu\|_2$. Likewise, $\|\mu_t - \mu_{t-1}\|_2 \mapsto \|R(\mu_t - \mu_{t-1})\|_2 = \|\mu_t - \mu_{t-1}\|_2$. Hence $\psi_t$ and any function (MLP + sigmoid) are invariant: $g_t' = g_t$.

**Proposition A.2 (Equivariance of gated location update).**    Under the same action, the update of gate is $E(2)$-equivariant.

*Proof.* Since $g_t$ is invariant (Lemma A.1), applying $g$ yields

$$\mu_t^{\text{new}} \mapsto (1 - g_t)(R\mu_t + t) + g_t(R\mu_{t-1} + t) \tag{15}$$

$$= R\big((1 - g_t)\mu_t + g_t\mu_{t-1}\big) + t \tag{16}$$

$$= R\mu_t^{\text{new}} + t. \tag{17}$$

$\square$

**Proposition A.3 (Equivariance of gated non-positional updates).** Assume gates $\gamma_t, \eta_t, \zeta_t$ are $E(2)$-invariant. Then the gated updates for $r, a, w$ above preserve the same group behavior as the ungated ones.

*Proof.*

1. Rotation: global action $r \mapsto r + \phi$ implies $(r_t + \phi) + \gamma_t \Delta r_t = (r_t + \gamma_t \Delta r_t) + \phi$, hence $SO(2)$-equivariant.

2. Scale in log-space is a scalar pair independent of $R$; its coupling to geometry is via $\Sigma(a, r) = R(r)\text{diag}(a^2)R(r)^\top$. Since $\log a_{t+1} = \log a_t + \eta_t \Delta \log a_t$, the resulting $\Sigma$ transforms as $R_\phi \Sigma R_\phi^\top$.

3. Weight $w$ is scalar and remains invariant.

$\square$

### A.2.2 GAUSSIAN ELLIPSOID

where $p_{ij} = [i, j]^\top$ is the coordinate vector of grid cell $(i, j)$, $A_k = \text{diag}(a_k)$ is the scaling matrix, and $R_k$ is the 2D rotation matrix given by

$$R_k = \begin{bmatrix} \cos r_k & -\sin r_k \\ \sin r_k & \cos r_k \end{bmatrix}. \tag{18}$$

## B  EXPERIMENT DETAILS

### B.1  DATASET DETAILS

We use two widely used weather datasets, NEXRAD (Department of Atmospheric Sciences, Texas A&M University & School of Meteorology, University of Oklahoma, 2021) and RainNet (Chen et al., 2022), for evaluation. *NEXRAD* consists of radar observations of major storms across the United States, including 6,255 frames of 3D radar observations with a grid size of $512 \times 512 \times 44$ and interval of 5 minites. Each voxel contains 7 radar features. We follow Wang et al. to split the dataset into training, validation, and testing sets with a ratio of 7:1:2 based on time. *RainNet* contains 62,400 pairs of high-quality low/high-resolution precipitation maps from 2002 to 2018 with an interval of 1 hour. The sizes of the high-resolution and low-resolution precipitation map are $624 \times 999$ and $208 \times 333$, respectively. We follow Chen et al. (2022) to split the dataset into training, validation, and testing sets with a ratio of 7:1:2 based on time.

### B.2  IMPLEMENTATION DETAILS

We train all the models on 8 NVIDIA H800 80GB GPUs and 8 NVIDIA 4090Ti 24GB GPUs. The network is trained using the Adam optimizer with a learning rate of 0.001 and a batch size of 16. The hidden dimension of all MLPs is set to 128. We use ReLU as the activation function and apply layer normalization after each MLP. The code will be released upon acceptance.

### B.3  METRIC DETAILS

Following previous works (Pathak et al., 2022; Lam et al., 2023; Chen et al., 2022; Wang et al.), we mainly employ Mean Squared Error (MSE), Mean Absolute Error (MAE), and Critical Success Index (CSI) to evaluate the forecasting performance. For data compression and super-resolution

tasks, we utilize the Peak Signal-to-Noise Ratio (PSNR) and the Structural Similarity Index Measure (SSIM) as evaluation metrics. For CSI, we follow Wang et al. to report the CsI at pooling scale $4 \times 4$, which relaxes the pixel-wise matching to evaluate the accuracy on neighborhood aggregations.

# C ADDITIONAL RESULTS

## C.1 ADDITIONAL ABLATION STUDY RESULTS

We also provide the ablation study results on each rotation angle in Table 6. One can observe that our equivariant design consistently outperforms the non-equivariant counterpart across all rotation angles, demonstrating the effectiveness of incorporating physical symmetries into the model architecture for weather forecasting tasks. Besides, the performance gap between the two models widens as the rotation angle increases, indicating that the equivariant model is more robust to larger transformations. Therefore, the equivariant design is crucial for achieving robust and accurate weather forecasting. Furthermore, other physical symmetries and dynamics can also be incorporated into the model to further enhance its performance and generalization ability, which we leave as future work.

Table 6: Ablation study on the effect of equivariance under different rotation angles. The arrows indicate whether higher ($\uparrow$) or lower ($\downarrow$) values are better.

| Equivariance | Angles | $e_\mu\downarrow$ | MAE $\downarrow$ | MSE $\downarrow$ | SSIM $\uparrow$ | LIPS $\downarrow$ |
|---|---|---|---|---|---|---|
| | 30 | 7.09 | 0.00261 | 0.0004 | 0.913 | 0.105 |
| | 60 | 12.50 | 0.00262 | 0.0004 | 0.913 | 0.105 |
| | 90 | 18.86 | 0.00265 | 0.0004 | 0.912 | 0.106 |
| | 120 | 23.92 | 0.00267 | 0.0004 | 0.911 | 0.107 |
| | 150 | 26.24 | 0.00270 | 0.0004 | 0.910 | 0.108 |
| | 180 | 26.49 | 0.00273 | 0.0004 | 0.910 | 0.109 |
| | 30 | 2.97 | 0.00166 | 0.0002 | 0.95 | 0.056 |
| | 60 | 3.12 | 0.00166 | 0.0002 | 0.95 | 0.056 |
| ✓ | 90 | 3.42 | 0.00167 | 0.0002 | 0.95 | 0.056 |
| | 120 | 3.86 | 0.00168 | 0.0002 | 0.95 | 0.056 |
| | 150 | 4.32 | 0.00169 | 0.0002 | 0.95 | 0.056 |
| | 180 | 4.95 | 0.00170 | 0.0002 | 0.95 | 0.057 |

## C.2 ADDITIONAL VISUAL RESULTS

**Equivariance** We randomly select three gaussian ellipsoids and visualize their evolution over a sequence of 8 time steps as shown in Fig. 7. One can observe that the Gaussian ellipsoids can effectively capture the motion and deformation of weather patterns over time.The predicted Gaussian ellipsoids by both our EGNN and the non-equivariant baseline can capture the general motion of the weather patterns (the first three rows). However, when the input sequence is rotated by $180°$, the predicted Gaussian ellipsoids by the non-equivariant baseline become distorted and misaligned with the actual weather patterns while our EGNN produces Gaussian ellipsoids that are correctly rotated and aligned with the weather patterns, demonstrating its ability to maintain geometric consistency under spatial transformations (the final three rows). Especially for the red ellipsoid, the non-equivariant baseline fails to capture its motion entirely, while our EGNN accurately tracks its trajectory. This result underscores the importance of incorporating physical symmetries into the model architecture for robust and reliable weather forecasting.

# D THE USE OF LARGE LANGUAGE MODELS (LLMS)

We use LLMs (GPT-5.0 and Gemini 2.5 pro) to polish our writing and check our grammar only.

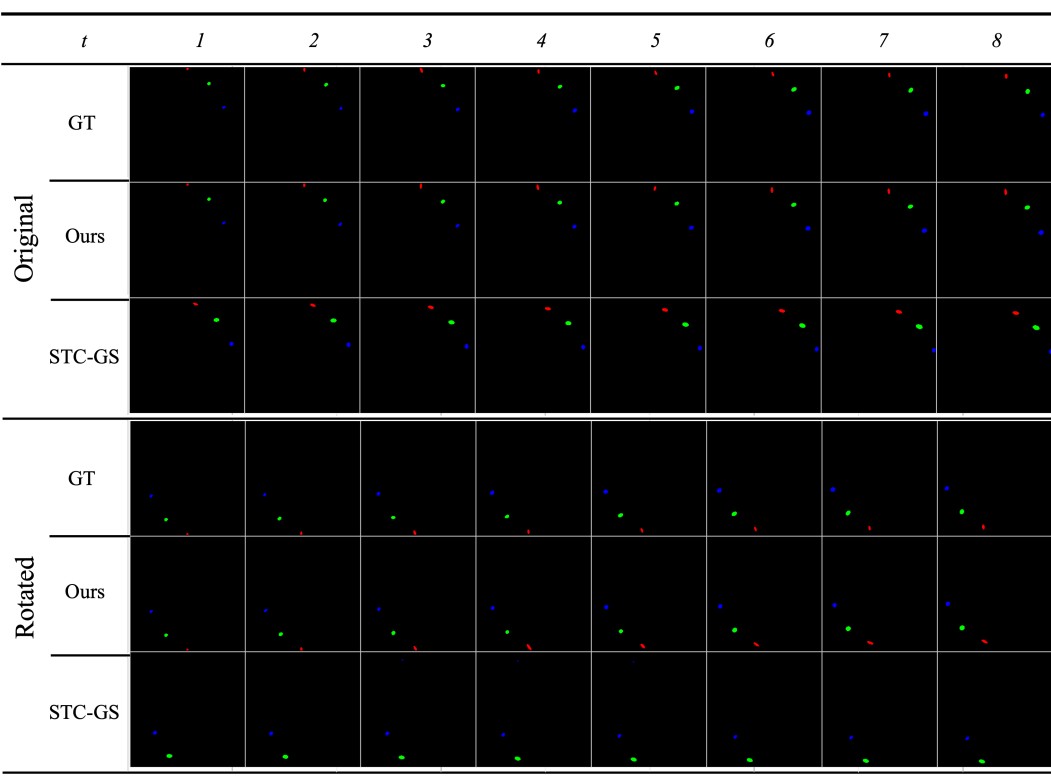

Figure 7: Prediction of Gaussian ellipsoid variation over time. First row: the ground truth ellipsoids. Second row: the predicted ellipsoids by our model. Third row: the predicted ellipsoids by the non-equivariant baseline. Fourth row: rotated ground truth ellipsoids. Fifth row: prediction with rotated input by our model. Sixth row: prediction with rotated input by non-equivariant baseline.

