# OpenReview forum: "EllipWeather: Gaussian Ellipsoid Representation for Weather Modeling"
_ICLR.cc/2026/Conference — Submitted to ICLR 2026_

### Official Review · Reviewer_YVmn · 2025-10-28

**Soundness:** 2
**Presentation:** 2
**Contribution:** 2
**Rating:** 2
**Confidence:** 5

**Summary:**

The paper proposes a weather forecasting paradigm by shifting from pixel-based grids to a compact, continuous representation using Gaussian ellipsoids. The authors introduce a hierarchical graph neural network (HGNN) for fitting ellipsoids from grid data and an equivariant graph neural network (EGNN) that respects rotational and translational symmetries for temporal prediction. Beyond forecasting, they also evaluate model performance on data compression and super-resolution.

**Strengths:**

1. The paper considers the symmetry of weather dynamics.
2. The authors highlight the massive data issues in weather forecasting.
3. Besides temporal forecasting, the authors also consider data compression and super-resolution problems.

**Weaknesses:**

* The introduction cites GraphCast and Aurora, which operate on terabyte-scale ERA5 datasets, to highlight data redundancy issues. However, the experiments use much smaller radar observations (NEXRAD) and precipitation maps (RainNet), undermining the relevance of these examples and potentially misleading readers about the method's applicability to large-scale global data.
* The core claim on line 52 that weather phenomena can be effectively modeled as Gaussian mixtures lacks in-depth explanation or empirical support. Decomposing complex weather into basis functions/distributions is challenging; prior work like Spherical Fourier Neural Operators (SFNO) attempted spherical harmonics representations but did not yield significant performance gains over baselines. As a central contribution, this assumption requires clearer validation.
* Why implement equivariance via EGNN rather than Spherical CNNs, which are more suited to grid-based data? The shift to graphs for Gaussian ellipsoids seems unnecessary, and the proposed method appears to merely apply existing EGNN principles. Moreover, equivariant models feel unintuitive for regional (non-global) datasets used here.
* If the data is regarded as graphs, the performance of related equivariant GNNs should be compared.
* The paper omits references to relevant works like Spherical Fourier Neural Operators (SFNO) and scaling spherical cnns.
* Table 1 lacks specification of the prediction time scale, but results suggest single-step forecasting. For robust validation and practical utility, multi-step or longer-term predictions should be evaluated.
* In Section 4.1.2, standard GNNs are not trained on rotated data, making their underperformance against EGNN expected. A fairer benchmark would compare the proposed method to GNNs trained on augmented, rotated datasets.
* The code is not provided.

**Questions:**

* An in-depth explanation or empirical support for that weather phenomena can be effectively modeled as Gaussian mixtures.
* Results on ERA5 datasets.
* Insights on equivariant GNN designs.
* More empirical results.

**Details Of Ethics Concerns:**

N/A.

---

### Official Review · Reviewer_quNA · 2025-10-29

**Soundness:** 3
**Presentation:** 2
**Contribution:** 2
**Rating:** 4
**Confidence:** 3

**Summary:**

This paper introduces EllipWeather, a novel framework for weather modeling that replaces traditional pixel-based representations with Gaussian ellipsoid representations. Each weather phenomenon is modeled as a set of Gaussian ellipsoids, allowing a compact, continuous, and resolution-independent description of atmospheric states. The authors design an E(2)-equivariant graph neural network (EGNN) that models the temporal evolution of these ellipsoids while preserving physical symmetries such as translation and rotation equivariance. Experiments on NEXRAD and RainNet datasets show state-of-the-art performance in forecasting accuracy, as well as strong results in data compression and super-resolution tasks.

**Strengths:**

Gaussian Representation Paradigm:
The Gaussian ellipsoid formulation is an physically meaningful alternative to pixel grids, effectively reducing redundancy and capturing the continuous nature of weather systems.

Strong Physical Inductive Bias:
The use of an equivariant GNN ensures consistency under spatial transformations, aligning well with the inherent symmetries of atmospheric dynamics.

Comprehensive Experimental Validation:
Extensive results on multiple datasets demonstrate clear improvements over strong baselines in both accuracy and structural similarity metrics.

**Weaknesses:**

* The paper lacks sufficient novelty in its proposed novel weather representation using Gaussian ellipsoids, as a similar idea has already been explored by Wang et al. [1], who represented weather patterns using Gaussian primitives.

* The authors claim that their method is “more compact and efficient than traditional pixel-based approaches,” yet they do not provide quantitative evidence of this efficiency improvement (e.g., computational cost or training time). Although Section 4.1.3 presents compression ratios and reconstruction quality under different numbers of ellipsoids, Table 1 does not report any computational comparisons with other models on the prediction task.

* The paper does not clarify how the number of Gaussian components was chosen for the prediction tasks, nor does it include an ablation study to analyze this hyperparameter’s influence.

* The practical significance of equivariance is not sufficiently demonstrated, and its applicational value appears limited. While the ablation studies show that EGNN performs better than non-equivariant models, the experimental setup (e.g., rotation by 180°) is too idealized and unrealistic for actual weather systems. Since equivariance ensures consistent predictions under spatial transformations (rotation, translation, etc.), it would be more convincing if the authors could connect this property to specific meteorological phenomena.

* Wang et al. [1] reported a 72× compression ratio on the NEXRAD dataset using STC-GS. In contrast, the proposed method achieves much lower compression, though potentially with higher reconstruction accuracy. A quantitative comparison with other data compression methods would help clarify this trade-off.

* The authors mention the ability to render ellipsoids onto arbitrary target grids without retraining (around lines 65–66), but there is no comparative evaluation of this capability on prediction tasks at different spatial resolutions.




















Reference:
---

[1] High-Dynamic Radar Sequence Prediction for Weather Nowcasting Using Spatiotemporal Coherent Gaussian Representation

**Questions:**

Are all experimentsconducted on 2D datasets, with no evaluation on 3D data? In contrast, Wang et al. [1] performed experiments on 3D datasets, which makes the current experimental validation appear somewhat limited in scope.

Possible typo: Line 344 — the paper states that a 2D Gaussian has 8 parameters (2 for location, 2 for scaling, 3 for rotation, and 1 for weight). However, this parameterization seems inconsistent and may need clarification or correction.

---

### Official Review · Reviewer_MJPL · 2025-10-31

**Soundness:** 2
**Presentation:** 2
**Contribution:** 2
**Rating:** 2
**Confidence:** 4

**Summary:**

This paper introduces EllipWeather, a novel framework for weather modeling that moves away from traditional grid-based representations. Instead of pixels, it represents weather phenomena as a collection of Gaussian ellipsoids, providing a continuous, compact, and resolution-agnostic description of the atmospheric state. To predict the evolution of these ellipsoids, the authors develop a custom E(2)-equivariant graph neural network (EGNN) that operates directly on the ellipsoid parameters, inherently respecting the rotational and translational symmetries of physical laws.

**Strengths:**

- Principled Equivariant Design: The development of the EGNN is a strong technical achievement. The careful handling of the different transformation properties of the ellipsoid parameters demonstrates a deep understanding of geometric deep learning and its application to physical systems.
- Multi-faceted Benefits: The framework is not just a better forecasting model; it is a fundamentally better representation. The demonstrated capabilities in compression and super-resolution are highly compelling and have significant practical value.

**Weaknesses:**

- Scalability to 3D and Multiple Fields: The current work focuses on a single 2D weather variable for simplicity. While the authors state that the method can be extended, scaling it to the full 3D atmospheric state with dozens of coupled variables is a non-trivial challenge. The number of ellipsoids required and the complexity of their interactions could grow significantly. A more detailed discussion of this scaling path would strengthen the paper.
- Ellipsoid Fitting Process: The paper proposes two methods for fitting ellipsoids to pixel data: an iterative optimization and a hierarchical GNN (HGNN). The HGNN is a clever idea, but its training and generalization capabilities could be explored further. It's unclear how robust this fitting process is to different types of weather phenomena or data sources.
- Comparison to Lagrangian Methods: The object-based approach has conceptual similarities to Lagrangian modeling frameworks in fluid dynamics, where the fluid is modeled as a set of moving particles. The paper would benefit from a discussion contextualizing EllipWeather with respect to these classical approaches and other particle-based GNN models.

**Questions:**

1.  Could you elaborate on the computational complexity of the EGNN, particularly how it scales with the number of ellipsoids, *K*? How is *K* chosen in practice, and is there a risk of it becoming a bottleneck for complex, multi-scale weather patterns?
2.  The HGNN for encoding pixel data into ellipsoids is an interesting contribution. How does the performance of the end-to-end system change when using the iterative fitting method versus the HGNN encoder during training and inference?
3.  How does the model handle the appearance and disappearance of weather phenomena (i.e., the creation or destruction of ellipsoids)? Does the current framework assume a constant number of ellipsoids, *K*, over time?

---

### Official Review · Reviewer_3ytQ · 2025-10-31

**Soundness:** 2
**Presentation:** 2
**Contribution:** 1
**Rating:** 2
**Confidence:** 4

**Summary:**

The paper proposes EllipWeather, which represents weather fields as collections of Gaussian ellipsoids and forecasts their temporal evolution with a group-equivariant GNN. The authors argue this object-based, symmetry-aware design is more compact than pixels and naturally supports compression and resolution-agnostic downscaling/super-resolution.

**Strengths:**

1. The object-based ellipsoid representation is clearly articulated and plausibly useful for compact storage.
2. The equivariance motivation and ablation are reasonable, with quantitative improvements.

**Weaknesses:**

1. Forecasting comparisons omit mainstream global models in WeatherBench 2 benchmark (Pangu, Graphcast, NeuralGCM, etc.) and validate on ERA5 data, which limitation its real-world application ability. Further, the real-word dynamicl system is not equivariance. You just conduct experiments on an ideal or easy-modeled dataset. I made some equivariance experiments; however, if I force the model to be equivariant, the results will be worser in real-world dataset. So, the equivariance is not suitable for real-world weather forecasting, it's just a good story.
2. Limited evidence of novelty beyond combining known components. This paper extends prior “primitive/gaussian” representations to weather and swaps a generic sequence model for an equivariant GNN.
3. Metrics include MAE/MSE/CSI/SSIM/LPIPS, but there is limited analysis of extreme events or calibration/physical fidelity (conservation, geostrophic balance, etc.).

**Questions:**

See Weaknesses

---

### Meta-Review · Area_Chair_fWwY · 2026-01-07

**Summary:**

There was no rebuttal posted.
The reviewers agree that the paper presents an interesting and clearly articulated idea, with a technically sound implementation and reasonable experimental results. However, the consensus is that the work does not meet the bar for acceptance in its current form. The majority recommendation is reject, with one reviewer viewing it as marginally below the acceptance threshold.

**Reviewer Concerns:**

Lack of novelty, experimental rigor, and relevance to realistic, large-scale weather forecasting. There was no rebuttal to address these concerns.

**Reviewer Scores:**

There will be no changes given there was no rebuttal

---

### Decision · Program_Chairs · 2026-01-26

Reject